# Finding and Visualizing Weaknesses of Deep Reinforcement Learning Agents

**Christian Rupprecht**[1]     **Cyril Ibrahim**[2]     **Christopher J. Pal**[2,3]

[1]*Visual Geometry Group, University of Oxford*
[2]*Element AI*
[3]*Polytechnique Montréal, Mila & Canada CIFAR AI Chair*

## Abstract

As deep reinforcement learning driven by visual perception becomes more widely used there is a growing need to better understand and probe the learned agents. Understanding the decision making process and its relationship to visual inputs can be very valuable to identify problems in learned behavior. However, this topic has been relatively under-explored in the research community. In this work we present a method for synthesizing visual inputs of interest for a trained agent. Such inputs or states could be situations in which specific actions are necessary. Further, critical states in which a very high or a very low reward can be achieved are often interesting to understand the situational awareness of the system as they can correspond to risky states. To this end, we learn a generative model over the state space of the environment and use its latent space to optimize a target function for the state of interest. In our experiments we show that this method can generate insights for a variety of environments and reinforcement learning methods. We explore results in the standard Atari benchmark games as well as in an autonomous driving simulator. Based on the efficiency with which we have been able to identify behavioural weaknesses with this technique, we believe this general approach could serve as an important tool for AI safety applications.

## 1 Introduction

Humans can naturally learn and perform well at a wide variety of tasks, driven by instinct and practice; more importantly, they are able to justify why they would take a certain action. Artificial agents should be equipped with the same capability, so that their decision making process is interpretable by researchers. Following the enormous success of deep learning in various domains, such as the application of convolutional neural networks (CNNs) to computer vision (LeCun et al., 1998; Krizhevsky et al., 2012; Long et al., 2015; Ren et al., 2015), a need for understanding and analyzing the trained models has arisen. Several such methods have been proposed and work well in this domain, for example for image classification (Simonyan et al., 2013; Zeiler & Fergus, 2014; Fong & Vedaldi, 2017), sequential models (Karpathy et al., 2016) or through attention (Xu et al., 2015).

Deep reinforcement learning (RL) agents also use CNNs to gain perception and learn policies directly from image sequences. However, little work has been so far done in analyzing RL networks. We found that directly applying common visualization techniques to RL agents often leads to poor results. In this paper, we present a novel technique to generate insightful visualizations for pre-trained agents.

Currently, the generalization capability of an agent is—in the best case—evaluated on a validation set of scenarios. However, this means that this validation set has to be carefully crafted to encompass as many potential failure cases as possible. As an example, consider the case of a self-driving agent, where it is near impossible to exhaustively model all interactions of the agent with other drivers, pedestrians, cyclists, weather conditions, even in simulation. Our goal is to extrapolate from the training scenes to novel states that induce a specified behavior in the agent.

In our work, we learn a generative model of the environment as an input to the agent. This allows us to probe the agent's behavior in novel states created by an optimization scheme to induce specific

actions in the agent. For example we could optimize for states in which the agent sees the only option as being to slam on the brakes; or states in which the agent expects to score exceptionally low. Visualizing such states allows to observe the agent's interaction with the environment in critical scenarios to understand its shortcomings. Furthermore, it is possible to generate states based on an objective function specified by the user. Lastly, our method does not affect and does not depend on the training of the agent and thus is applicable to a wide variety of reinforcement learning algorithms.

Our contributions are:

1. We introduce a series of objectives to quantify different forms of *interestingness* and danger of states for RL agents.

2. We evaluate our algorithm on 50 Atari games and a driving simulator, and compare performance across three different reinforcement learning algorithms.

3. We quantitatively evaluate parts of our model in a comprehensive loss study (Tab. 1) and analyze generalization though a pixel level analysis of synthesized unseen states (Tab. 2).

4. An extensive supplement shows additional comprehensive visualizations on 50 Atari games.

We will describe our method before we will discuss relevant related work from the literature.

## 2 METHODS

We will first introduce the notation and definitions that will be used through out the remainder of the paper. We formulate the reinforcement learning problem as a discounted, infinite horizon Markov decision process $(\mathcal{S}, \mathcal{A}, \gamma, P, r)$, where at every time step $t$ the agent finds itself in a state $s_t \in \mathcal{S}$ and chooses an action $a_t \in \mathcal{A}$ following its policy $\pi_\theta(a|s_t)$. Then the environment transitions from state $s_t$ to state $s_{t+1}$ given the model $P(s_{t+1}|s_t, a_t)$. Our goal is to visualize RL agents given a user-defined objective function, without adding constraints on the optimization process of the agent itself, i.e. assuming that we are given a previously trained agent with fixed parameters $\theta$.

We approach visualization via a generative model over the state space $\mathcal{S}$ and synthesize states that lead to an interesting, user-specified behavior of the agent. This could be, for instance, states in which the agent expresses high uncertainty regarding which action to take or states in which it sees no good way out. This approach is fundamentally different than saliency-based methods as they always need an input for the test-set on which the saliency maps can be computed. The generative model constrains the optimization of states to induce specific agent behavior.

### 2.1 STATE MODEL

Often in feature visualization for CNNs, an image is optimized starting from random noise. However, we found this formulation too unconstrained, often ending up in local minima or fooling examples (Figure 3a). To constrain the optimization problem we learn a generative model on a set $\mathcal{S}$ of states generated by the given agent that is acting in the environment. The model is inspired by variational autoencoders (VAEs) (Kingma & Welling, 2013) and consists of an encoder $f(s) = (f_\mu(s), f_\sigma(s)) \in \mathbb{R}^{2 \times n}$ that maps inputs to a Gaussian distribution in latent space and a decoder $g(z(s)) = \hat{s}$ that reconstructs the input. $z(s) = f_\mu(s) + f_\sigma(s) \odot \epsilon$ is a sample from the predicted distribution, that is obtained via the reparametrization trick, where $\epsilon$ is sampled from $\mathcal{N}(0, I_n)$. The training of our generator has three objectives. First, we want the generated samples to be close to the manifold of valid states $s$. To avoid fooling examples, the samples should also induce correct behavior in the agent and lastly, sampling states needs to be efficient. We encode these goals in three corresponding loss terms.

$$\mathcal{L}(s) = \mathcal{L}_p(s) + \eta \mathcal{L}_a(s) + \mathcal{KL}(f(s), \mathcal{N}(0, I_n)), \tag{1}$$

The role of $\mathcal{L}_p(s)$ is to ensure that the reconstruction $g(z(s))$ is close to the input $s$ such that $\| g(z(s)) - s \|_2^2$ is minimized. We observe that in the typical reinforcement learning benchmarks, such as Atari games, small details—e.g. the ball in Pong or Breakout—are often critical for the decision making of the agent. However, a typical VAE model tends to yield blurry samples that are not able to capture such details. To address this issue, we model the reconstruction error $\mathcal{L}_p(s)$ with an *attentive loss* term, which leverages the saliency of the agent to put focus on critical regions of the

reconstruction. The saliency maps are computed by guided backpropagation of the policy's gradient with respect to the state.

$$\mathcal{L}_p(s) = \sum_{i=1}^{d} \frac{(g(z(s))_i - s_i)^2 \, |\nabla \pi(s)_i|}{\sum_{j=1}^{d} |\nabla \pi(s)_j|}, \tag{2}$$

where $i$ and $j$ iterate over all $d$ pixels in the image/gradient. As discussed earlier, gradient based reconstruction methods might not be ideal for explaining a CNN's reasoning process (Kindermans et al., 2017a). Here however, we only use it to focus the reconstruction on salient regions of the agent and do not use it to explain the agent's behavior for which these methods are ideally suited. This approach puts emphasis on details (salient regions) when training the generative model.

Since we are interested in the actions of the agent on synthesized states, the second objective $\mathcal{L}_a(s)$ is used to model the *perception of the agent*:

$$\mathcal{L}_a(s) = \| A(s) - A(g(z(s))) \|_2^2, \tag{3}$$

where $A$ is a generic formulation of the output of the agent. For a DQN for example, $\pi(s) = \max_a A(s)_a$, i.e. the final action is the one with the maximal $Q$-value. This term encourages the reconstructions to be interpreted by the agent the same way as the original inputs $s$. The last term $\mathcal{KL}(f(s), \mathcal{N}(0, I_n))$ ensures that the distribution predicted by the encoder $f$ stays close to a Gaussian distribution. This allows us to initialize the optimization with a reasonable random vector later and forms the basis of a regularizer. Thus, after training, the model approximates the distribution of states $p(s)$ by sampling $z$ directly from $\mathcal{N}(0, I_n)$. The generative model ($f$ and $g$) is trained with (1). We will then use the generator inside an optimization scheme to generate state samples that satisfy a user defined target objective.

## 2.2 Sampling States of Interest

Training a generator with the objective function of Equation 1 allows us to sample states that are not only visually close to the real ones, but which the agent can also interpret and act upon as if they were states from a real environment.

We can further exploit this property and formulate an energy optimization scheme to generate samples that satisfy a specified objective. The energy operates on the latent space $x = (x_\mu, x_\sigma)$ of the generator and is defined as the sum of a target function $T$ on agent's policy and a regularizer $R$

$$E(x) = T(\pi(g(x_\mu + x_\sigma \odot \epsilon))) + \alpha R(x). \tag{4}$$

The target function can be defined freely by the user and depends on the agent that is being visualized. For a DQN, one could for example define $T$ as the Q-value of a certain action, e.g. pressing the brakes of a car. In section 2.3, we show several examples of targets that are interesting to analyze. The regularizer $R$ can again be chosen as the KL divergence between $x$ and the normal distribution:

$$R(x) = \mathcal{KL}(x, \mathcal{N}(0, I_n)), \tag{5}$$

forcing the samples that are drawn from the distribution $x$ to be close to the Gaussian distribution that the generator was trained with. We can optimize equation 4 with gradient descent on $x$ as detailed in algorithm 1.

## 2.3 Target Functions

Depending on the agent, one can define several interesting target functions $T$ – we present and explore seven below, which we refer to as: $T^+$, $T^-$, $T^\pm$, $S^+$, $S^-$, and action maximization. For a DQN the previously discussed action maximization is interesting to find situations in which the agent assigns a high value to a certain action e.g. $T_{left}(s) = -A_{left}(s)$. Other states of interest are those to which the agent assigns a low (or high) value for all possible actions $A(s) = q = (q_1, \ldots, q_m)$. Consequently, one can optimize towards a low Q-value for the highest valued action with the following objective:

$$T^-(q) = \frac{\sum_{i=1}^{m} q_i e^{\beta q_i}}{\sum_{k=1}^{m} e^{\beta q_k}}, \tag{6}$$

---

**Algorithm 1** Optimize $x$ for target $T$

---
1: **Input:** Target objective $T$, step size $\lambda$, regularizer weight $\alpha$, trained generator $g$
2: **Output:** $x$
3: $x_\mu \leftarrow \mathbf{0}$
4: $x_\sigma \leftarrow I_n$
5: **while** not converged **do**
6:      $\epsilon \leftarrow$ sample from $\mathcal{N}(0, I_n)$
7:      $z \leftarrow x_\mu + x_\sigma \odot \epsilon$                          $\triangleright$ sample $z$ from $x$
8:      $s \leftarrow g(z)$                               $\triangleright$ generate state $s$ using $g$
9:      $x_\mu \leftarrow x_\mu - \lambda \frac{\partial}{\partial x_\mu} \left( T(\pi(s)) + \alpha R(x) \right)$     $\triangleright$ gradient step using (4) and (5)
10:     $x_\sigma \leftarrow x_\sigma - \lambda \frac{\partial}{\partial x_\sigma} \left( T(\pi(s)) + \alpha R(x) \right)$
11: **end while**

---

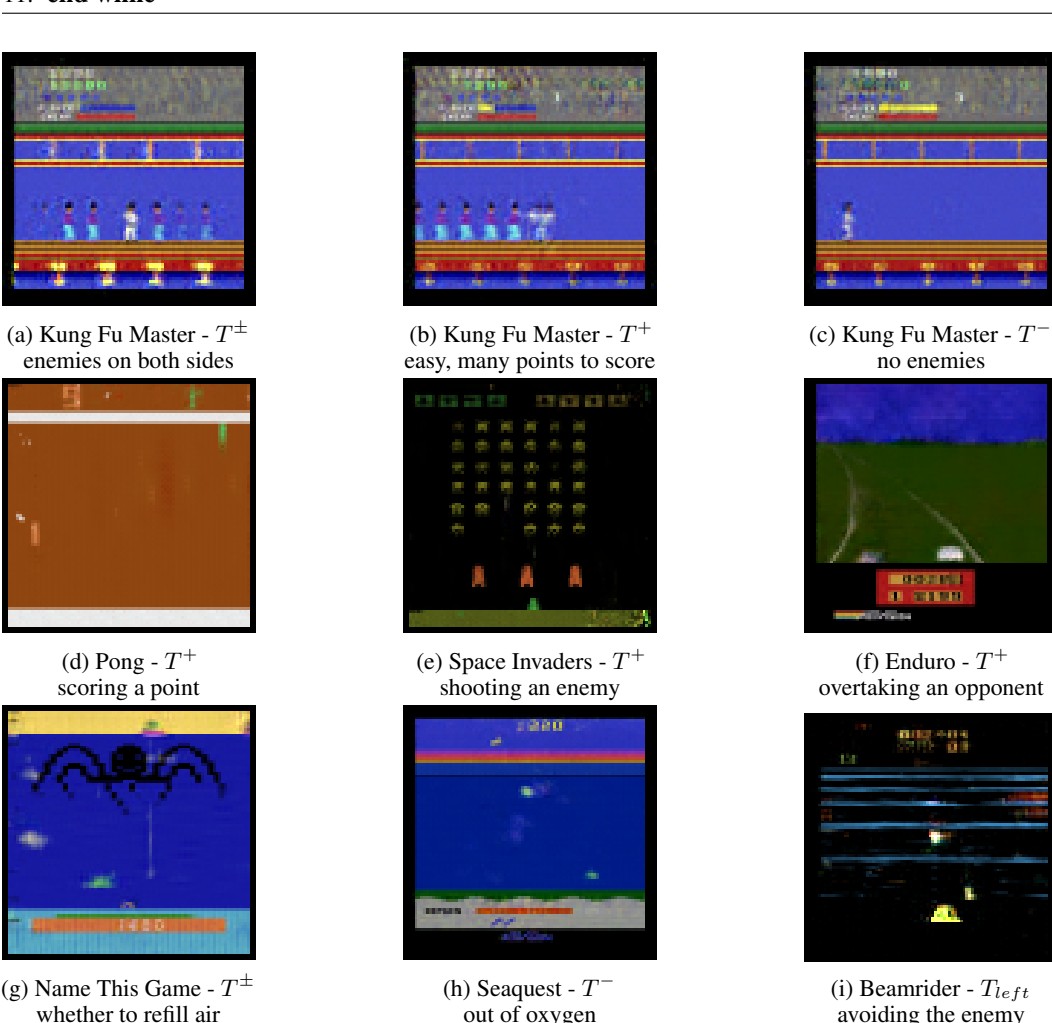

(a) Kung Fu Master - $T^\pm$
enemies on both sides

(b) Kung Fu Master - $T^+$
easy, many points to score

(c) Kung Fu Master - $T^-$
no enemies

(d) Pong - $T^+$
scoring a point

(e) Space Invaders - $T^+$
shooting an enemy

(f) Enduro - $T^+$
overtaking an opponent

(g) Name This Game - $T^\pm$
whether to refill air

(h) Seaquest - $T^-$
out of oxygen

(i) Beamrider - $T_{left}$
avoiding the enemy

Figure 1: **Qualitative Results:** Visualization of different target functions (Sec. 2.3). $T^+$ generates high reward and $T^-$ low reward states; $T^\pm$ generates states in which one action is highly beneficial and another is bad. For a long list of results, with over 50 Atari games, please see the appendix.

where $\beta > 0$ controls the sharpness of the soft maximum formulation. Analogously, one can maximize the lowest $Q$-value with $T^+(q) = -T^-(-q)$. We can also optimize for interesting situations in which one action is of very high value and another is of very low value by defining

$$T^\pm(q) = T^-(q) - T^+(q). \tag{7}$$

Finally, we can optimize for overall good states with $S^+(q) = \sum_{i=1}^m q_i$ and overall bad states with $S^- = S^-(-q)$.

# 3 Related Work

We divide prior work into two parts. First we discuss the large body of visualization techniques developed primarily for image recognition, followed by related efforts in reinforcement learning.

## 3.1 Feature Visualization

In the field of computer vision, there is a growing body of literature on visualizing features and neuron activations of CNNs. As outlined in (Grün et al., 2016), we differentiate between *saliency methods*, that highlight decision-relevant regions given an input image, methods that synthesize an image (pre-image) that fulfills a certain criterion, such as *activation maximization* (Erhan et al., 2009) or *input reconstruction*, and methods that are perturbation-based, i.e. they quantify how input modification affects the output of the model.

### 3.1.1 Saliency Methods

Saliency methods use the gradient of a prediction at the input image to estimate importance of pixels. Following gradient magnitude heatmaps (Simonyan et al., 2013) and class activation mapping (Zhou et al., 2016), more sophisticated methods such as (Mahendran & Vedaldi, 2016; Selvaraju et al., 2016) have been developed (Zintgraf et al., 2017) distinguish between regions in favor and regions speaking against the current prediction. (Sundararajan et al., 2017) distinguish between sensitivity and implementation invariance.

An interesting observation is that such methods seem to generate believable saliency maps even for networks with random weights (Adebayo et al., 2018a). (Kindermans et al., 2017b) show that saliency methods do not produce analytically correct explanations for linear models and further reliability issues are discussed in (Adebayo et al., 2018b; Hooker et al., 2018; Kindermans et al., 2017a).

### 3.1.2 Perturbation Methods

Perturbation methods modify a given input to understand the importance of individual image regions. (Zeiler & Fergus, 2014) slide an occluding rectangle across the image and measure the change in the prediction, which results in a heatmap of importance for each occluded region. This technique is revisited by (Fong & Vedaldi, 2017) who introduce blurring/noise in the image, instead of a rectangular occluder, and iteratively find a minimal perturbation mask that reduces the classifier's score, while (Dabkowski & Gal, 2017) train a network for masking salient regions.

### 3.1.3 Input Reconstruction

As our method synthesizes inputs to the agent, the most closely related work includes input reconstruction techniques. (Long et al., 2014) reconstruct an image from nearest neighbor patches in feature space. (Mahendran & Vedaldi, 2015) propose to reconstruct images by inverting CNN representations, while (Dosovitskiy & Brox, 2015) learn to reconstruct the input from its encoding.

When maximizing the activation of a specific class or neuron, regularization is crucial because the optimization procedure—starting from a random noise image and maximizing an output—is vastly under-constrained and often tends to generate fooling examples that fall outside the manifold of realistic images (Nguyen et al., 2015). In (Mahendran & Vedaldi, 2016) total variation (TV) is used for regularization, while (Baust et al., 2018) propose an update scheme based on Sobolev gradients. In (Nguyen et al., 2015) Gaussian filters are used to blur the pre-image or the update computed in every iteration. Since there are usually multiple input families that excite a neuron, (Nguyen et al., 2016c) propose an optimization scheme for the distillation of these clusters. More variations of regularization can be found in (Olah et al., 2017; 2018). Instead of regularization, (Nguyen et al., 2016a;b) use a denoising autoencoder to reconstruct pre-images for image classification.

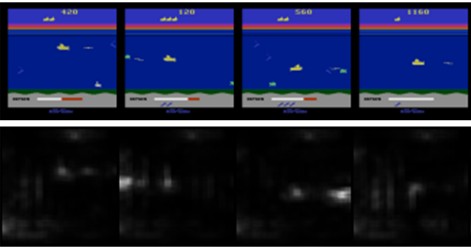

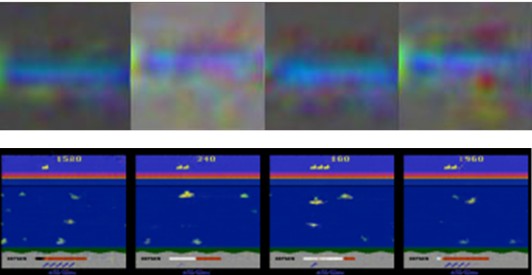

Figure 2: **Weight Visualization.** We visualize the weighting (second row) of the reconstruction loss from Equation 2 for eight randomly drawn samples (first row) of the dataset. Most weight lies on the player's submarine and close enemies, supporting their importance for the decision making.

Figure 3: **Comparison with activation maximization.** The visual features learned by the agents are not complex enough to reconstruct typical frames from the game via activation maximization (top). This problem is mitigated in our method by learning a low-dimensional embedding of games states (bottom).

## 3.2 EXPLANATIONS FOR REINFORCEMENT LEARNING

In deep reinforcement learning however, feature visualization is to date relatively unexplored. (Zahavy et al., 2016) apply t-SNE (Maaten & Hinton, 2008) on the last layer of a deep Q-network (DQN) to cluster states of behavior of the agent. (Mnih et al., 2016) also use t-SNE embeddings for visualization, while (Greydanus et al., 2017) examine how the current state affects the policy in a vision-based approach using *saliency methods*. (Wang et al., 2016) use saliency methods from (Simonyan et al., 2013) to visualize the value and advantage function of their dueling Q-network. (Huang et al., 2018) finds critical states of an agent based on the entropy of the output of a policy. (Uesato et al., 2018) adversarially search for initial states of the agent and environment that make it fail. Interestingly, we could not find prior work using *activation maximization* methods for visualization. In our experiments we show that the typical methods fail in the case of RL networks and generate images far outside the manifold of valid game states, even with all typical forms of regularization. This is additionally motivated by the insight that agents are vulnerable to adversarial states and can be manipulated even by other agents in the environment (Gleave et al., 2019).

## 4 EXPERIMENTS

In this section we thoroughly evaluate and analyze our method on Atari games (Bellemare et al., 2013) using the OpenAI Gym (Brockman et al., 2016) and a driving simulator. We present qualitative results for three different reinforcement learning algorithms, show examples on how the method helps finding flaws in an agent, analyze the loss contributions and compare to previous techniques.

### 4.1 IMPLEMENTATION DETAILS

In all our experiments we use the same factors to balance the loss terms in Equation 6: $\lambda = 10^{-4}$ for the KL divergence and $\eta = 10^{-3}$ for the agent perception loss. The generator is trained on $10,000$ frames (using the agent and an $\epsilon$-greedy policy with $\epsilon = 0.1$). Optimization is done with Adam (Kingma & Ba, 2015) with a learning rate of $10^{-3}$ and a batch size of 16 for 2000 epochs. Training takes approximately four hours on a Titan Xp. Our generator uses a latent space of 100 dimensions, and consists of four encoder stages comprised of a $3 \times 3$ convolution with stride 2, batch-normalization (Ioffe & Szegedy, 2015) and ReLU layer. The starting number of filters is 32 and is doubled at every stage. A fully connected layer is used for mean and log-variance prediction. Decoding is inversely symmetric to encoding, using deconvolutions and halving the number of channels at each of the four steps.

For the experiments on the Atari games we train a double DQN (Wang et al., 2016) for two million steps with a reward discount factor of 0.95. The input size is $84 \times 84$ pixels. Therefore, our generator performs up-sampling by factors of 2, up to a $128 \times 128$ output, which is then center cropped to $84 \times 84$ pixels. The agents are trained on grayscale images, for better visual quality however, our

generator is trained with color frames and convert to grayscale using a differentiable, weighted sum of the color channels. In the interest of reproducibility we will make the visualization code available.

## 4.2 Visualizations On Atari Games

In Figure 1, we show qualitative results from various Atari games using different target functions $T$, as described in Section 2.3. From these images we can validate that the general visualizations that are obtained from the method are of good quality and can be interpreted by a human. $T^+$ generates generally high value states independent of a specific action (first row of Figure 1), while $T^-$ generates low reward situations, such as close before losing the game in Seaquest (Figure 1.e) or when there are no points to score (Figure 1.i). Critical situations can be found by maximizing the difference between lowest and highest estimated Q-value with $T^\pm$. In those cases, there is clearly a right and a wrong action to take. In Name This Game (Figure 1.d) this occurs when close to the air refill pipe, which prevents suffocating under water; in Kung Fu Master when there are enemies coming from both sides (Figure 1.g), the order of attack is critical, especially since the health of the agent is low (yellow/blue bar on top). An example of maximizing the value of a single action (similar to maximizing the confidence of a class when visualizing image classification CNNs) can be seen in (Figure 1.f) where the agent sees moving left and avoiding the enemy as the best choice of action.

## 4.3 ACKTR

To show that this visualization technique generalizes over different RL algorithms, we also visualize ACKTR (Wu et al., 2017). We use the code and pretrained RL models from a public repository (Kostrikov, 2018) and train our generative model with the same hyperparameters as above and without any modifications on the agent. We present the $T^\pm$ objective for the ACKTR agent in Figure 4 to visualize states with both high and low rewards, for example low oxygen (surviving vs. suffocating) or close proximity to enemies (earning points vs. dying).

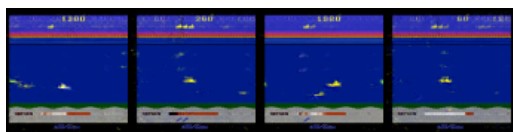

Figure 4: **Seaquest with ACKTR.** The objective is $T^\pm$ for situations that can be rewarding but also have a low scoring outcome. The generated states show low oxygen or close proximity to enemies.

Compared to the DQN visualizations the ACKTR visualizations, are almost identical in terms of image quality and interpretability. This supports the notion that our proposed approach is independent of the specific RL algorithm.

## 4.4 Interpretation of Visualizations

Analyzing the visualizations on Seaquest, we make an interesting observation. When maximizing the $Q$-value for the actions, in many samples we see a low or very low oxygen meter. In these cases the submarine would need to ascend to the surface to avoid suffocation. Although the up action is the only sensible choice in this case, we also obtain visualized low oxygen states for all other actions. This implies that the agent has not understood the importance of resurfacing when the oxygen is low. We then run several roll outs of the agent and see that the major cause of death is indeed suffocation and not collision with enemies. This shows the impact of visualization, as we are able to understand a flaw of the agent. Although it would be possible to identify this flawed behavior directly by analyzing the $10,000$ frames of training data for our generator, it is significantly easier to review a handful of samples from our method. Further, as the generator is a generative model, we can synthesize states that are not part of its training set.

## 4.5 Ablation Studies (Loss Terms)

In this section we analyze the three loss terms of our generative model. The human perceptual loss is weighted by the (guided) gradient magnitude of the agent in Equation 2. In Figure 2 we visualize this mask for a DQN agent for random frames from the dataset. The masks are blurred with an averaging filter of kernel size 5. We observe that guided backpropagation results in precise saliency maps focusing on player and enemies that then focus the reconstructions on what is important for the agent.

Table 1: **Loss Study.** We compare the performance of the original agent with the agent operating on reconstructed frames instead. The original performance is an upper bound for the score of the same agent which is operating on reconstructions instead. Shown are average scores over 20 runs.

|  | Agent | VAE | $\mathcal{L}_p$ only | Ours (full) |
| --- | --- | --- | --- | --- |
| Pong | 14 | -8 | 4 | 14 |
| Atlantis | 108 | 95 | 98 | 109 |
| Q*bert | 64 | 26 | 28 | 31 |

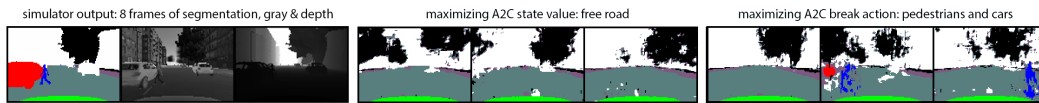

Figure 5: **Driving simulator.** Input frame sample on the left and then two target function visualizations obtained by our method. For each objective we show three *random* samples. For simplicity we only show the first frame of segmentation instead of the whole state (8 frames).

To study the influence of the loss terms we perform an experiment in which we evaluate the agent not on the real frames but on their reconstructions. If the reconstructed frames are perfect, the agent with *generator goggles* achieves the same score as the original agent. We can use this metric to understand the quantitative influence of the loss terms. In Pong, the ball is the most important visual aspect of the game for decision making.

In Table 1 we see that the VAE baseline scores much lower than our model. This can be explained as follows. Since the ball is very small, it is mostly ignored by the reconstruction loss of a VAE. The contribution of one pixel to the overall loss is negligible and the VAE never focuses on reconstructing the important part of the image. Our formulation is built to regain the original performance of the agent, by reweighing the loss on perceptually salient regions of the agent. Overall, we see that our method always improves over the baseline but does not always match the original performance.

## 4.6 COMPARISON WITH ACTIVATION MAXIMIZATION

For image classification tasks, activation maximization works well when optimizing the pre-image directly (Mahendran & Vedaldi, 2015; Baust et al., 2018). However we find that for reinforcement learning, the features learned by the network are not complex enough to reconstruct meaningful pre-images, even with sophisticated regularization techniques. The pre-image converges to a *fooling example* maximizing the class but being far away from the manifold of states of the environment.

In Figure 3.a we compare our results with the reconstructions generated using the method of (Baust et al., 2018) for a DQN agent. We obtain similarly bad pre-images with TV-regularization (Mahendran & Vedaldi, 2016), Gaussian blurring (Nguyen et al., 2015) and other regularization tricks such as random jitter, rotations, scaling and cropping (Olah et al., 2017). This shows that it is not possible to directly apply common techniques for visualizing RL agents and explains why a learned regularization from our generator is needed to produce meaningful examples.

## 4.7 EXPERIMENTS WITH A DRIVING SIMULATOR

Driving a car is a continuous control task set within a much more complex environment than Atari games. To explore the behavior of our proposed technique in this setting we have created a 3D driving simulation environment and trained an A2C agent maximizing speed while avoiding pedestrians that are crossing the road.

In our first set of experiments we trained an A2C agent to maximize speed while avoiding swerving and pedestrians that are crossing the road. The input to the agent are eight temporal frames comprised of depth, semantic segmentation and a gray-scale image (Figure 5). With this experiment we visualize three random samples for two target functions. The moving car and person categories, appear most prominently when probing the agent for the break action. However, we are also able to identify a

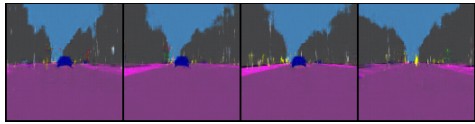

Figure 6: **Driving simulator.** Samples for the $T^-$ objective of an agent trained in the *reasonable pedestrians* environment. From these samples one can infer that the agent is aware of traffic lights (red) and other cars (blue) but has very likely not understood the severity of hitting pedestrians (yellow). Deploying this agent in the *distracted pedestrians* environment shows that the agent indeed collides with people that cross the road in front of the agent.

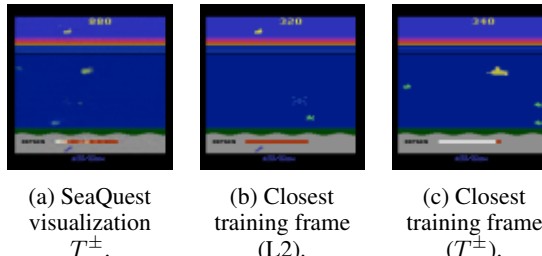

(a) SeaQuest visualization $T^{\pm}$.

(b) Closest training frame (L2).

(c) Closest training frame ($T^{\pm}$).

Figure 7: **Generating novel states.** We show a frame generated by our method under the $T^{\pm}$ objective and retrieve the closest frame from the training set using L2 distance and the objective function. Both frames are very different, showing that the method is able to generate novel states. For a quantitative evaluation, please see Tab. 2.

Table 2: **Synthesizing unseen states.** We compare generated samples to their closest neighbor in the training set and compute the percentage of pixels whose values differ by at least 25%, e.g. 73% of the synthesized samples differ in more than 20% pixels in comparison to their closest training sample.

| difference | $> 10\%$ | $> 20\%$ | $> 30\%$ | $> 50\%$ | $> 70\%$ |
|---|---|---|---|---|---|
| samples | 99% | 73% | 16% | 1% | 0% |

flaw: unnecessary braking on empty roads as shown in the left most image of the right most block of three frames. Inappropriate breaking is a well known issue in this problem domain.

In a second set of experiments, we use our simulator to build two custom environments and validate that we can identify problematic behavior in the agent. The agent is trained with four temporal semantic segmentation frames ($128 \times 128$ pixels) as input (Figure 6). We train the agent in a "*reasonable pedestrians*" environment, where pedestrians cross the road carefully, when no car is coming or at traffic lights. With these choices, we model data collected in the real world, where it is unlikely that people unexpectedly run in front of the car. We visualize states in which the agent expects a low future return ($T^-$ objective) in Figure 6. It shows that the agent is aware of other cars, traffic lights and intersections. However, there are no generated states in which the car is about to collide with a person, meaning that the agent does not recognize the criticality of pedestrians. To verify our suspicion, we test this agent in a "*distracted pedestrians*" environment where people cross the road looking at their phones without paying attention to approaching cars. We find that the agent does indeed run over humans. With this experiment, we show that our visualization technique can identify biases in the training data just by critically analyzing the sampled frames.

## 4.8 NOVEL STATES

To be able to generate novel states is useful, since it allows the method to model new scenarios that were not accounted for during training of the agent. This allows the user to identify potential problems without the need to include every possible permutation of situations in the simulator or real-world data collection.

While one could simply examine the experience replay buffer to find scenarios of interest, our approach allows unseen scenarios to be synthesized. To quantitatively evaluate the assertion that our generator is capable of generating novel states, we sample states and compare them to their closest frame in the training set under an MSE metric. We count a pixel as different if the relative difference in a channel exceeds 25% and report the histogram in Table 2. The results show that there are very few samples that are very close to the training data. On average a generated state is different in 25% of the pixels, which is high, considering the overall common layout of the road, buildings and sky.

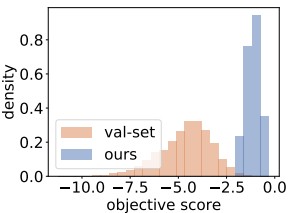 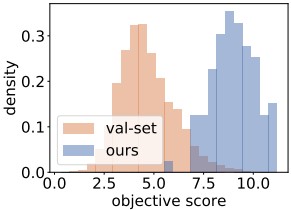 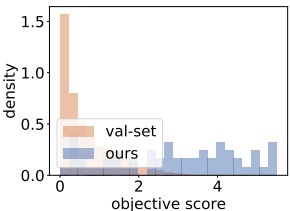

(a) Histogram objective $T^-$   (b) Histogram objective $T^+$   (c) Histogram objective $T^\pm$

Figure 8: **Objective Score Distribution.** We compare the distribution of values of different target functions ($T^-$ (a), $T^+$ (b) and $T^\pm$ (c)) between states from the validation set and those generated by our method. It is clear that the distribution of critical states differs and partially lies outside the maximum values of the val-set. This means these states are novel states, with higher *criticalness*-score.

We examine these results visually for Atari SeaQuest in Fig. 7, where we show a generated frame and the L2-closest frame from the training set additional to the closest frame in the training set based on the objective function. Retrieval with L2 is, as usual not very meaningful on images since the actual interesting parts of the images are dominated by the background. Thus we have also included a retrieval experiment based on the objective score which shows the submarine in a similar gameplay situation but with different enemy locations. The results in Tab. 2 and Fig. 7 confirm that the method is able to generate unseen states and does not overfit to the training set.

To understand whether one could simply sample states from the replay buffer or a validation set, we compare the distribution of the target function values in Fig. 8 (on the example of the "Boxing" environment). It is clear that our method is able to generate states with a much more targeted distribution towards the objective. In all three cases, we are even able to generate states that score higher target values than any state in the validation set. This confirms that we are able to generate targeted and novel critical states and thus cannot be obtained from the validation set alone.

## 5 DISCUSSION AND CONCLUSIONS

We have presented a method to synthesize inputs to deep reinforcement learning agents based on generative modeling of the environment and user-defined objective functions. The agent perception loss helps the reconstructions to focus on regions of the current state that are important to the agent and avoid generating fooling examples. Training the generator to produce states that the agent perceives as those from the real environment enables optimizing its latent space to sample states of interest. Please consult the appendix for more extensive visualization experiments.

We believe that understanding and visualizing agent behavior in safety critical situations is a crucial step towards creating safer and more robust agents using reinforcement learning. We have found that the methods explored here can indeed help accelerate the detection of problematic situations for a given learned agent. For our car simulation experiments we have focused upon the identification of weaknesses in constructed scenarios; however, we see great potential to apply these techniques to much more complex simulation environments with less obvious safety critical weaknesses.

## ACKNOWLEDGMENTS

We would like to thank Iro Laina, Alexandre Piché, Simon Ramstedt, Evan Racah and Adrien Ali Taiga for helpful discussions and proofreading. We thank the Open Philanthropy project for supporting C.R. while he was an intern at Mila, where this work began.

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

APPENDIX

To show an unbiased and wide variety of results, in the following, we will show four random samples generated by our method for a DQN agent trained on many of the Atari benchmark environments. We show visualizations optimized for a meaningful objective for each game (e.g. not optimizing for unused buttons). All examples were generated with the same hyperparameter settings.

Please note that for some games better settings can be found. Some generators on visually more complex games would benefit from longer training to generate sharper images. Our method is able to generate reasonable images even when the DQN was unable to learn a meaningful policy such as for *Montezuma's revenge* (Fig. 40). We show two additional objectives maximizing/minimizing the expected reward of the state under a random action: $S^+(q) = \sum_{i=1}^{m} q_i$ and $S^-(q) = -S^+(q)$. Results in alphabetical order and best viewed in color.

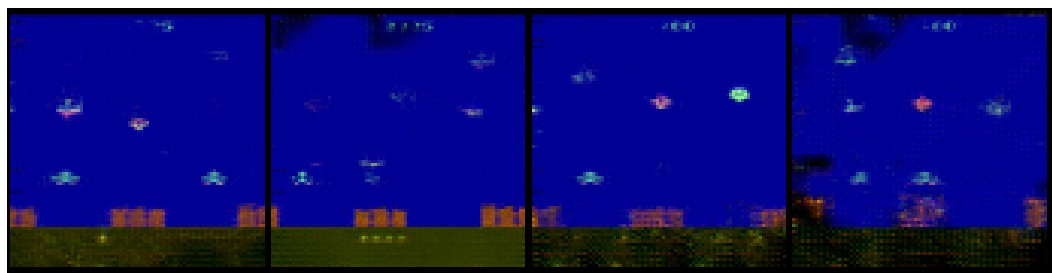

Figure 9: **Air Raid.** Target function: $S^+$.

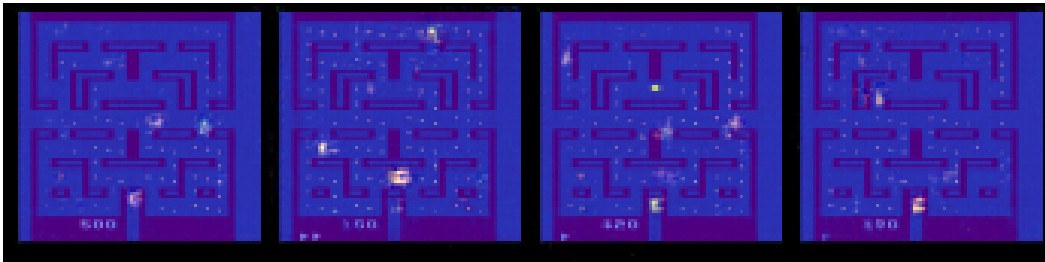

Figure 10: **Alien.** Target function: right.

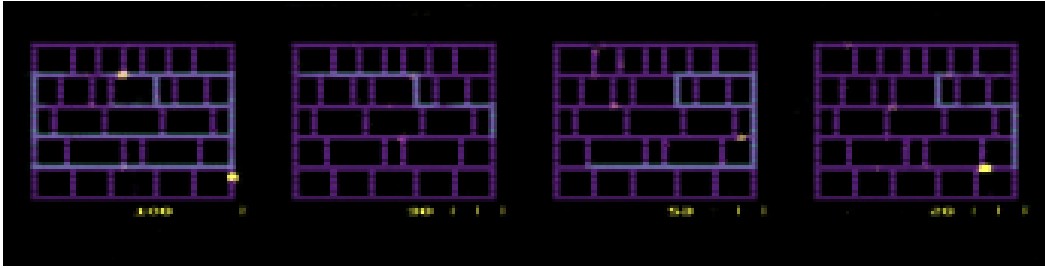

Figure 11: **Amidar.** Target function: up.

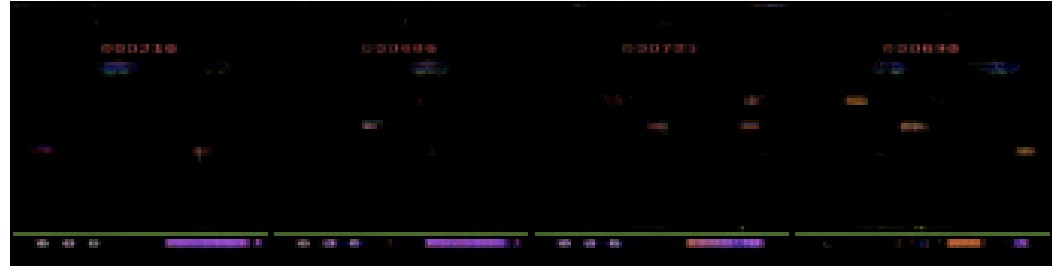

Figure 12: **Assault.** Target function: $S^-$.

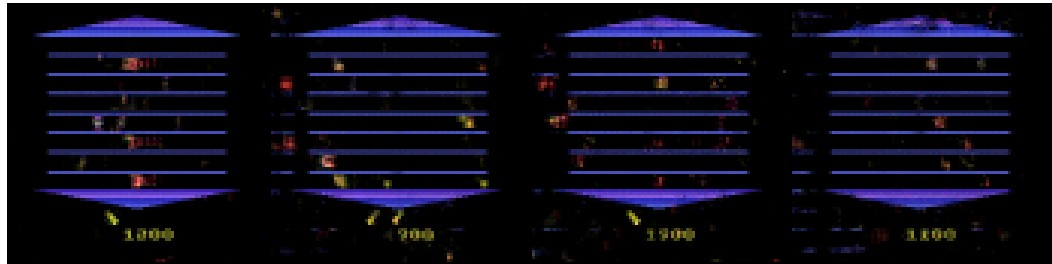

Figure 13: **Asterix.** Target function: $T^-$.

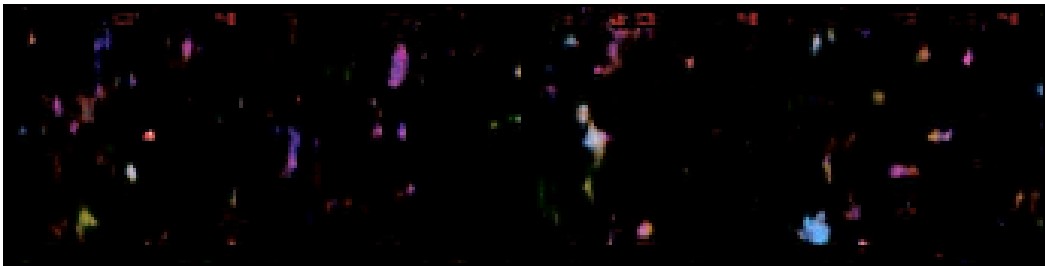

Figure 14: **Asteroids.** Target function: up-fire.

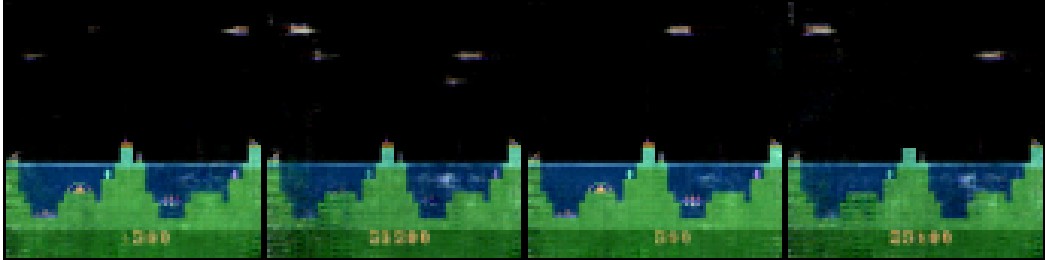

Figure 15: **Atlantis.** Target function: $T^+$.

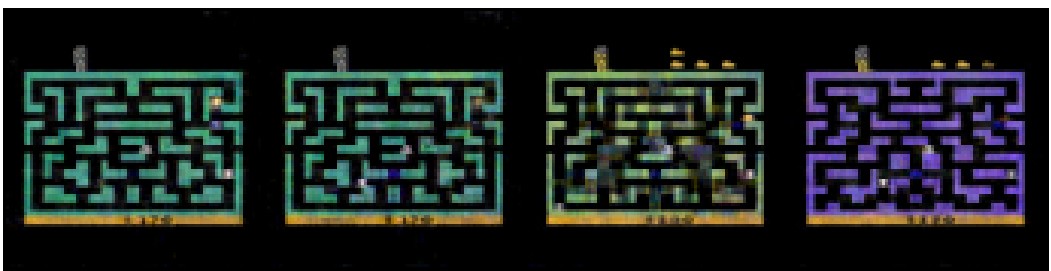

Figure 16: **Bank Heist.** Target function: $T^+$.

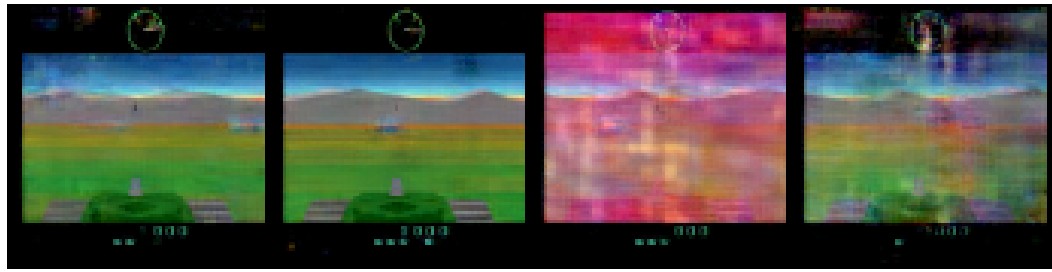

Figure 17: **Battlezone.** Target function: $T^-$.

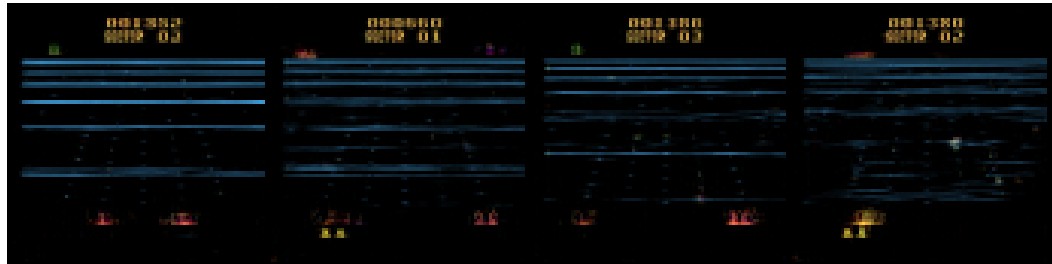

Figure 18: **Beamrider.** Target function: $T^+$.

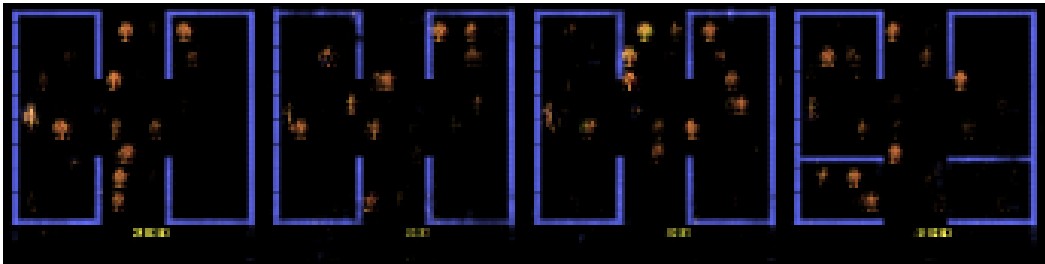

Figure 19: **Berzerk.** Target function: $S^+$.

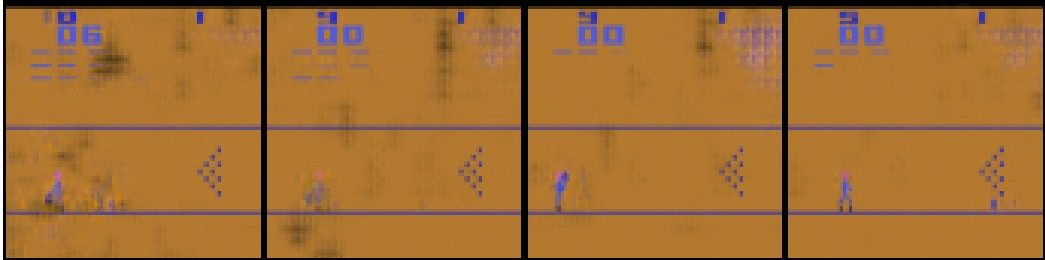

Figure 20: **Bowling.** Target function: $S^+$.

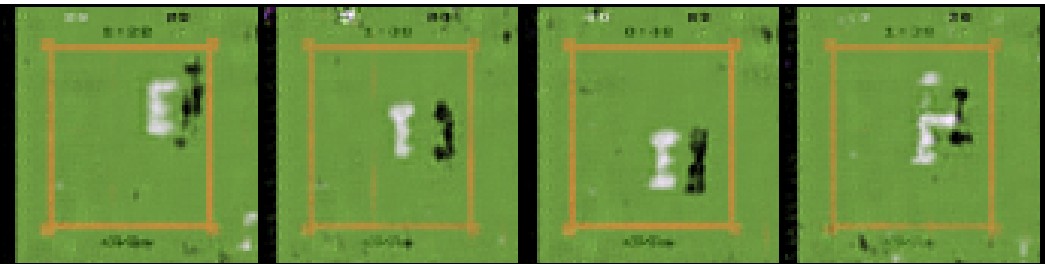

Figure 21: **Boxing.** Target function: $S^+$.

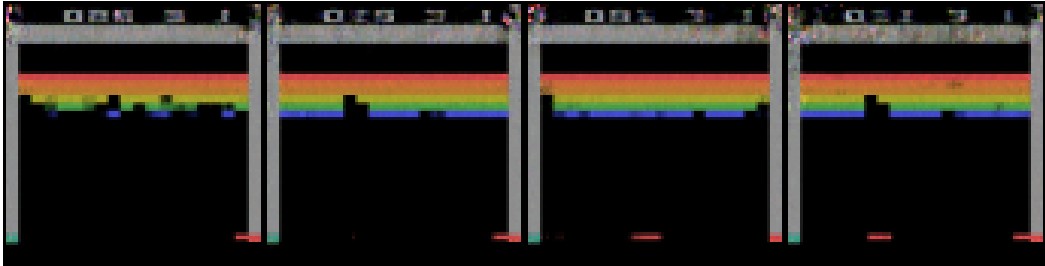

Figure 22: **Breakout.** Target function: $T^-$.

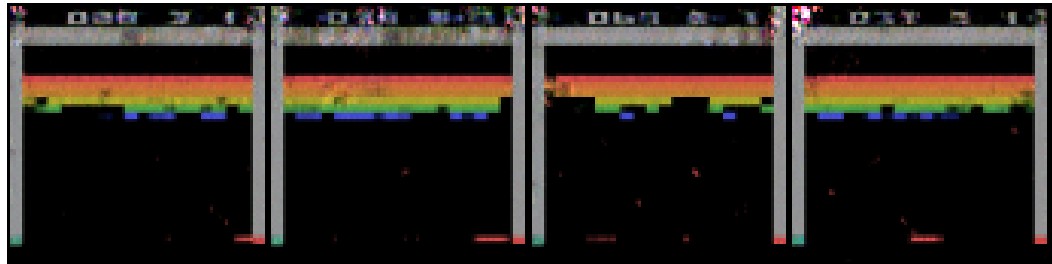

Figure 23: **Breakout.** Target function: Left.

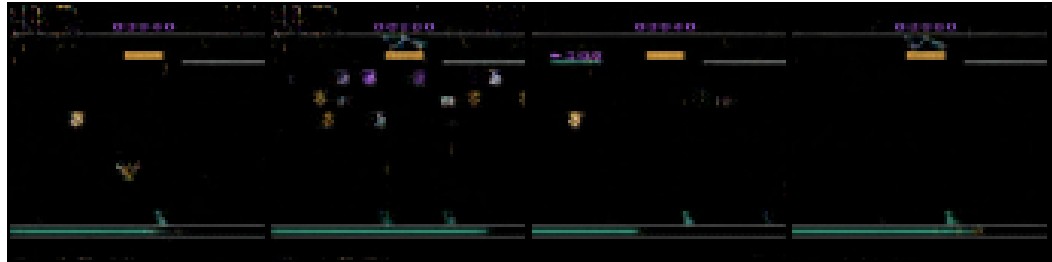

Figure 24: **Carnival.** Target function: right.

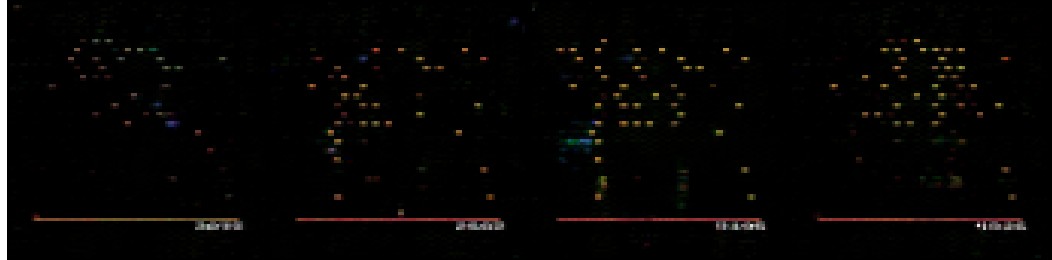

Figure 25: **Centipede.** Target function: $T^{\pm}$.

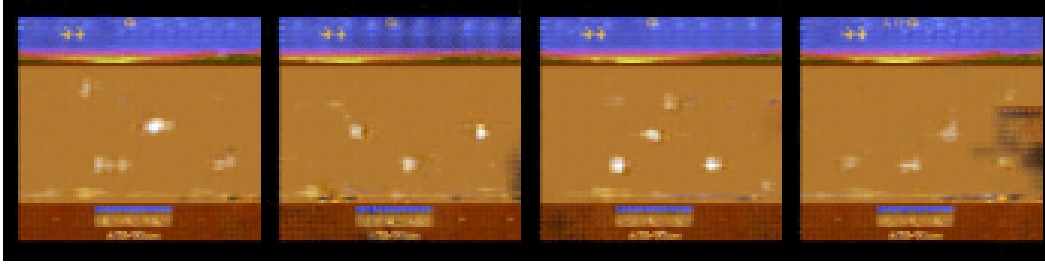

Figure 26: **Chopper Command.** Target function: $S^+$.

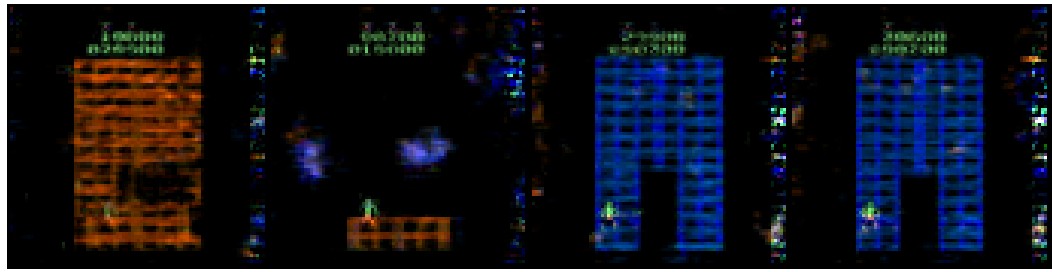

Figure 27: **Crazy Climber.** Target function: $T^-$.

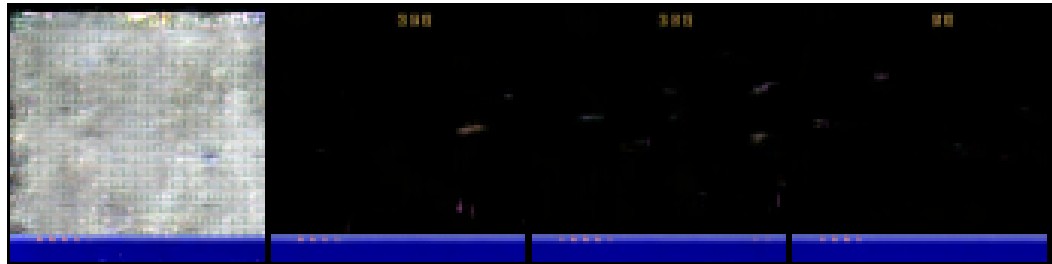

Figure 28: **Demon Attack.** Target function: $T^+$.

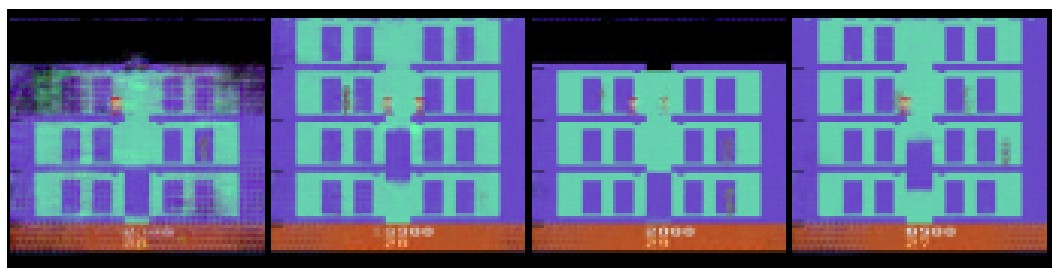

Figure 29: **Elevator Action.** Target function: no-op.

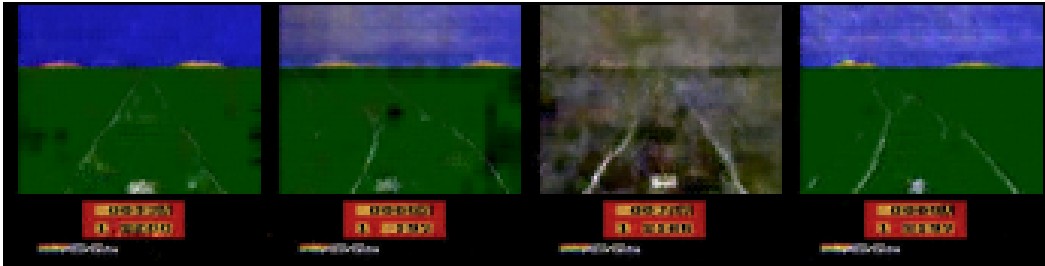

Figure 30: **Enduro.** Target function: $S^+$.

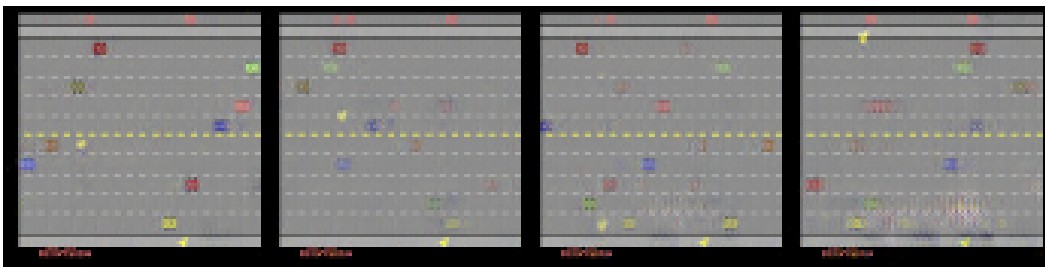

Figure 31: **Freeway.** Target function: $T^+$.

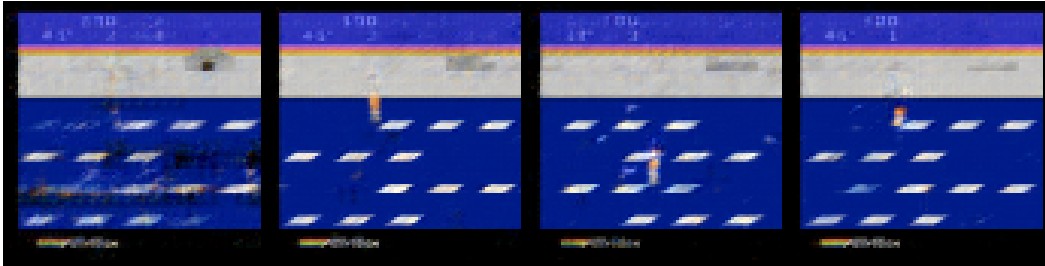

Figure 32: **Frostbite.** Target function: no-op.

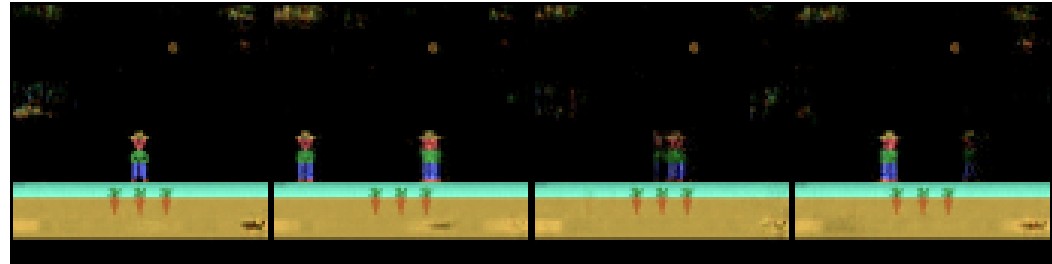

Figure 33: **Gopher.** Target function: $S^-$.

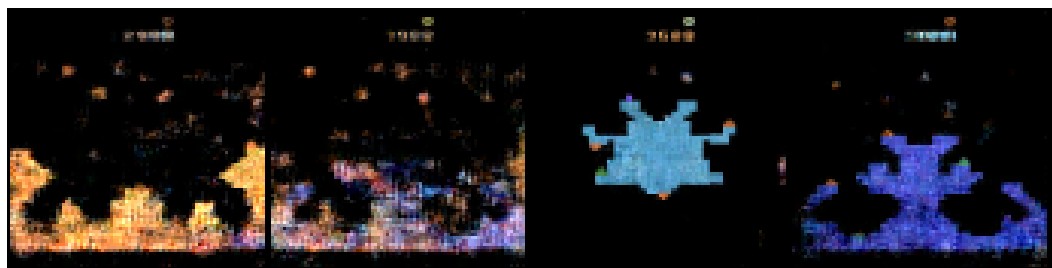

Figure 34: **Gravitar.** Target function: $T^\pm$.

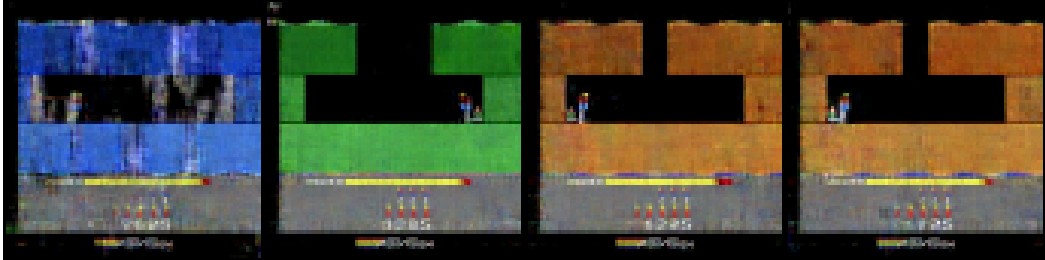

Figure 35: **Hero.** Target function: $S^+$.

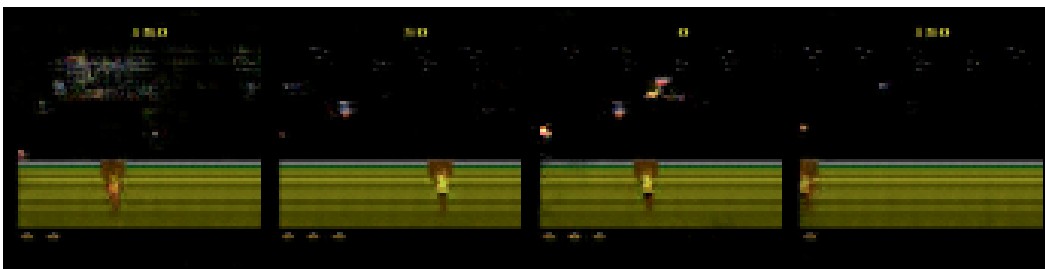

Figure 36: **JamesBond.** Target function: $S^+$.

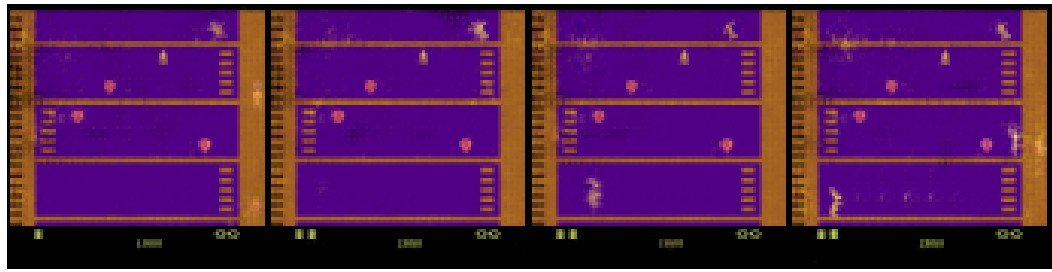

Figure 37: **Kangaroo.** Target function: $S^-$.

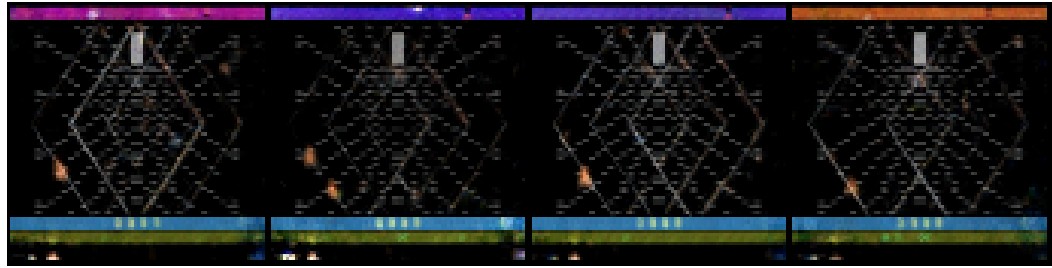

Figure 38: **Krull.** Target function: fire.

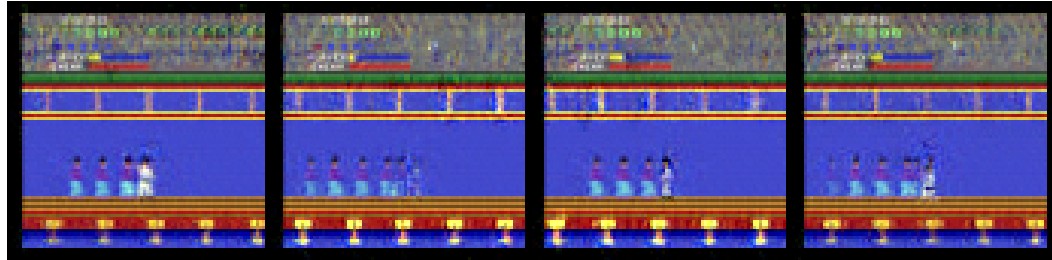

Figure 39: **Kung Fu Master.** Target function: up.

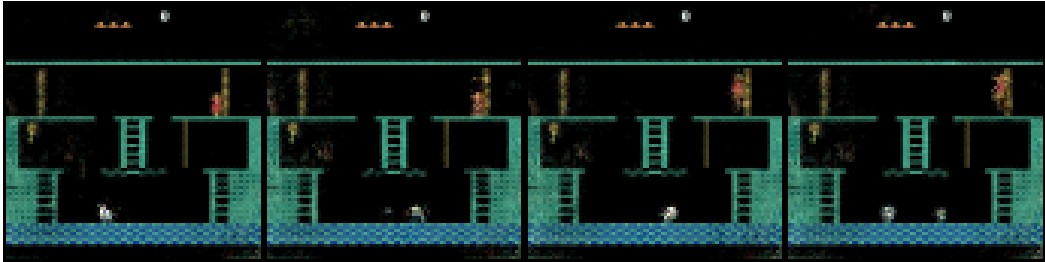

Figure 40: **Montezuma's Revenge.** Target function: $T^-$.

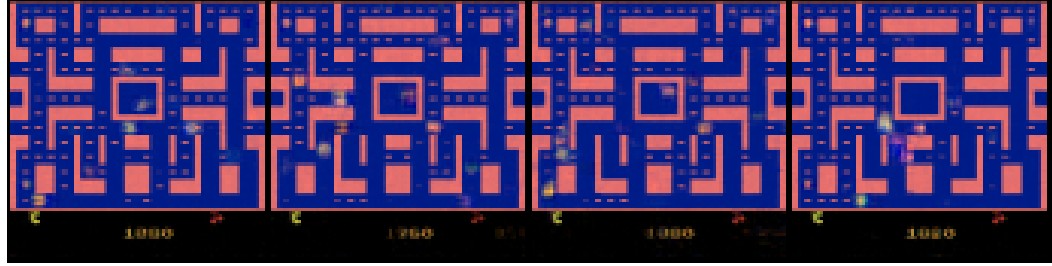

Figure 41: **Ms. Pacman.** Target function: no-op.

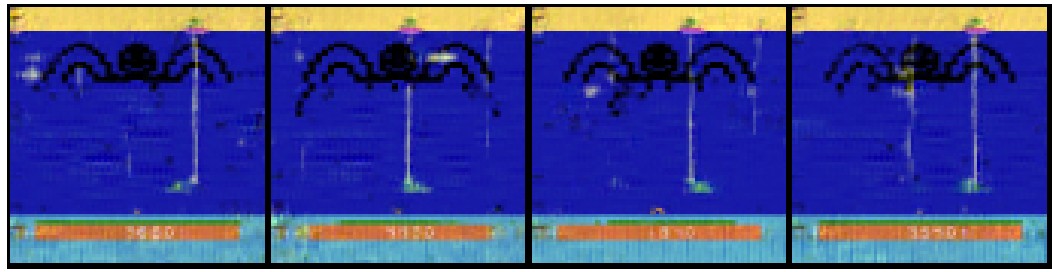

Figure 42: **Name This Game.** Target function: $T^{\pm}$.

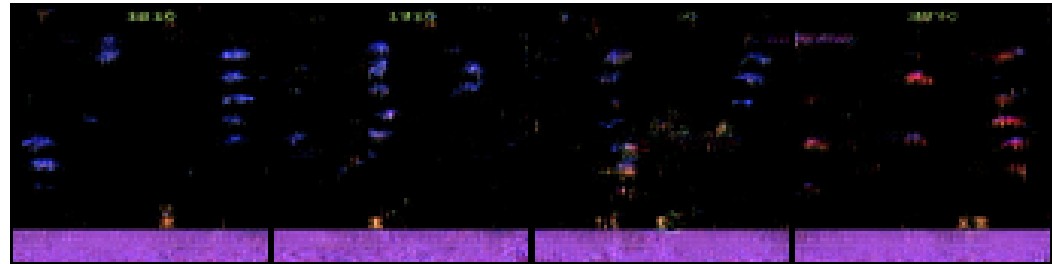

Figure 43: **Phoenix.** Target function: $T^{\pm}$.

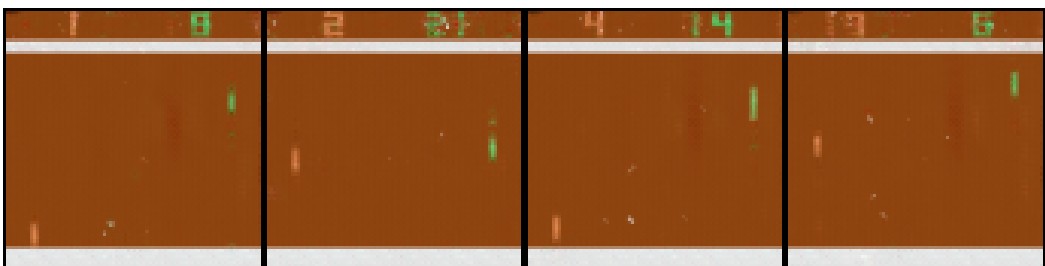

Figure 44: **Pong.** Target function: no-op.

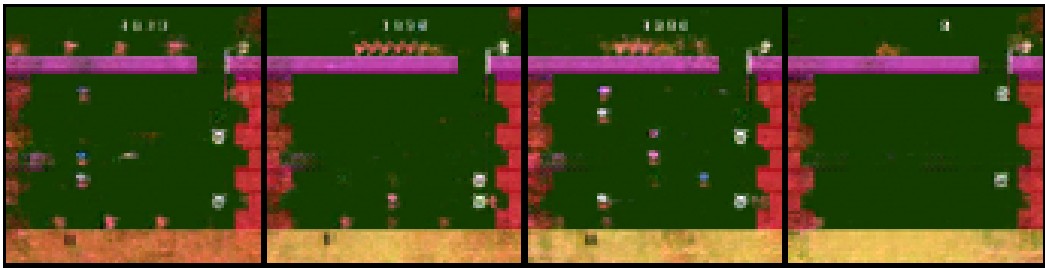

Figure 45: **Pooyan.** Target function: $S^{-}$.

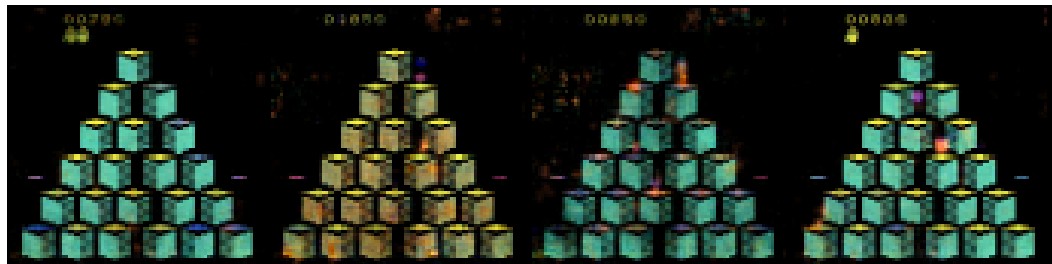

Figure 46: **Q-Bert.** Target function: left.

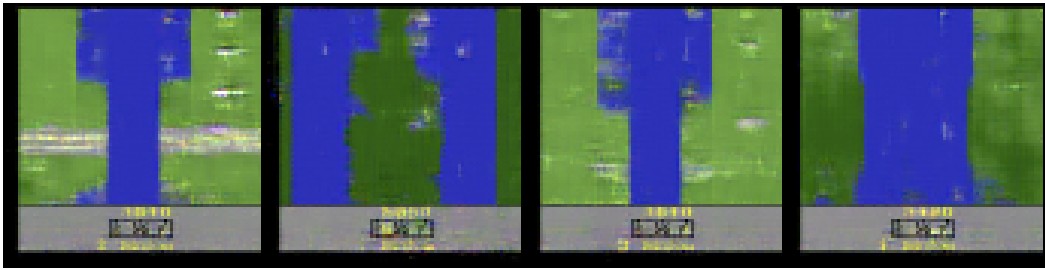

Figure 47: **River Raid.** Target function: $T^+$.

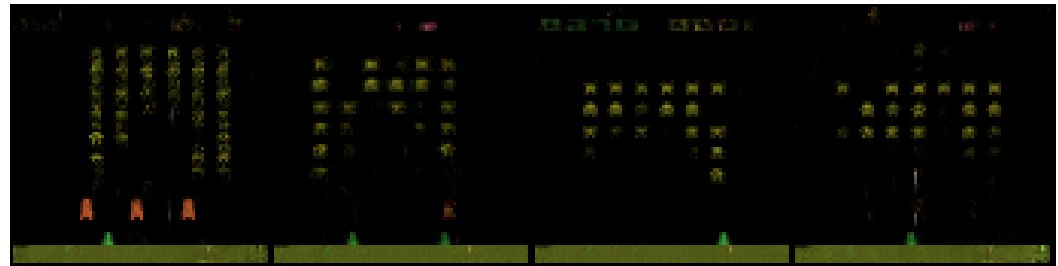

Figure 48: **Space Invaders.** Target function: left.

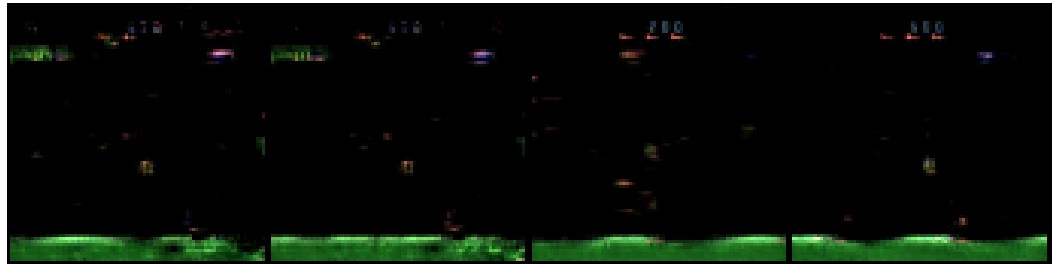

Figure 49: **Star Gunner.** Target function: $T^{\pm}$.

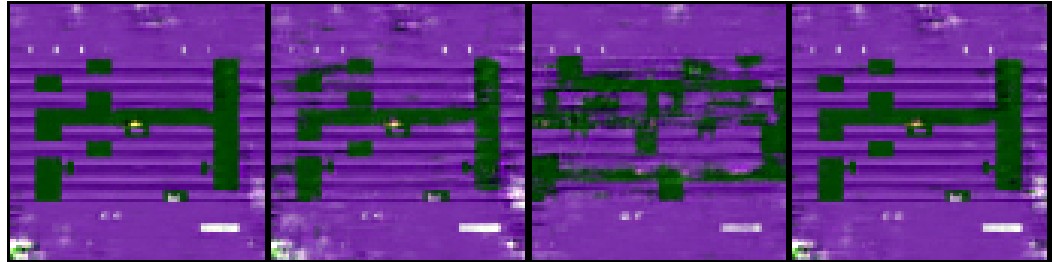

Figure 50: **Tutankham.** Target function: no-op.

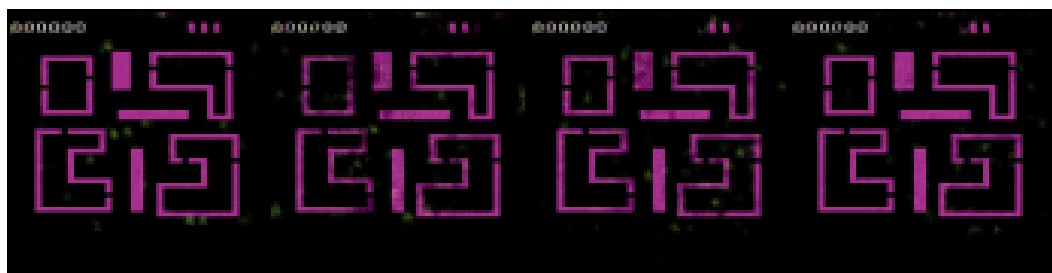

Figure 51: **Venture.** Target function: $S^+$.

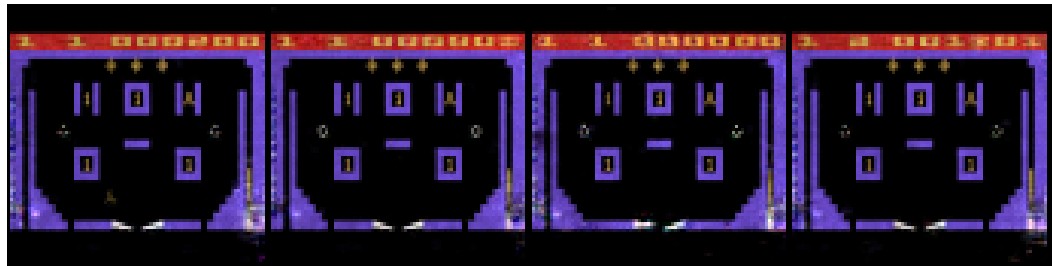

Figure 52: **Video Pinball.** Target function: $T^-$.

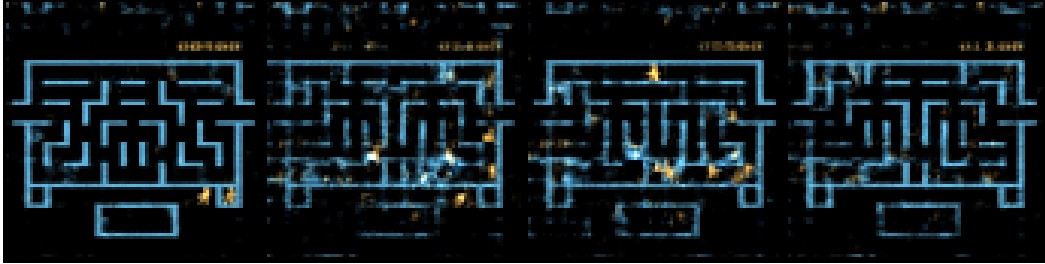

Figure 53: **Wizard Of Wor.** Target function: left.

