# OpenReview forum: "Finding and Visualizing Weaknesses of Deep Reinforcement Learning Agents"
_ICLR.cc/2020/Conference — Accept (Poster)_

### Official Review · AnonReviewer1 · 2019-10-17
**Official Blind Review #1**

**Rating:** 3

**Review:**

Summary

This paper proposes a generative technique to sample "interesting" states useful for analyzing the behavior of deep reinforcement learning agents. In this context, the concept of "interesting" is defined via user-specific target functions, e.g. states that arise as a consequence of taking specific actions (such as actions associated with high or low Q-values for example). The approach is evaluated in the Atari domain and in an autonomous driving simulator. Results are mainly presented as visualizations of interesting states that are described verbally.

Quality

The quality of the submission is extremely low. The optimization objectives chosen by the authors seem very ad hoc to me and how the motivation relates to the objectives is hard to comprehend (see my Clarity section). The experimental results have very low quality as well---results are mainly depicted as images with a verbal explanation.

Clarity

The clarity of the paper is extremely poor. While I do conceptually understand Section 2.1, I have a hard time linking it precisely to Section 2.2. Just some examples regarding lack in clarity:
- What is z in Section 2.1?
- How do the objectives in Section 2.1 and Section 2.2 relate to each other, i.e. how does the algorithm operate? Some pseudocode would be really helpful here.
- What are the target functions S^+, S^- and S^\pm in Section 2.3?
- What is the difference in the KL-regularizer mentioned in the text below Equation (3) and in Equation (5)?
- In the text above Equation (2), it is mentioned that a squared reconstruction loss is insensitive to small elements in the image that have a huge impact on the reward. While this is true, I don't see how the multiplicative policy gradient norm term in Equation (2), as proposed by the authors, is addressing this issue. The proposed modification puts emphasis on states where the norm of the policy gradient is high, which is different from putting emphasis on specific regions in the image. I guess the intention would be to do an element-wise multiplication of the squared loss vector and the absolute value policy gradient vector before collapsing to a scalar, or something similar?
In general, I found the entire writing from Section 3 onward a bit wordy and I do not think that nine pages are required to deliver the message of the paper in its current form.

Originality

The idea of visualizing states that reveal interesting insights about an agent's behavior based on a user-defined target function sounds interesting. But I have not worked in interpretability of agent behavior, which is why I leave the assessment of the originality to the other reviewers and the area chair.

Significance

If the results of the paper were backed up with some proper scientific metrics other than verbally explaining images, there might be some significance in the paper.

Update

After the authors' response, I am currently not inclined to change my score. While I do agree that the paper proposes an interesting idea, the technical presentation of the work is simply too poor and not convincing at this stage. Here are a few points:

- A variational autoencoder works as follows. There is a generative model over latent variables z and observed variables s, consisting of a prior for z and a likelihood for s conditioned on z. The prior over z can be e.g. a normal distribution denoted as N(z|\mu_prior, \Sigma_prior). Then the likelihood (decoder) can be e.g. a deep neural net that maps z to a normal distribution of s denoted as N(s|\mu_likelihood(z, \theta), \Sigma_likelihood(z, \theta)) where \theta refers to the decoder's neural network weights. Furthermore, there is a recognition model that approximates the posterior over z given s (encoder)---this can also be e.g. a deep neural net that maps s to a normal distribution in z denoted as N(z|\mu_posteriorapprox(s, \psi), \Sigma_posteriorapprox(s, \psi)) where \psi refers to the encoder's neural network weights.

- The reparameterization trick is not required for the technical explanation of the involved random variables and how they relate to each other. It is merely an optimization trick to establish a functional dependency between a random variable and the parameters of its distribution (e.g. mean and covariance in the Gaussian case).

- To be specific about your updated paper. The notation you chose for the encoder f(s) = (\mu, \sigma) is confusing because it hides the dependency on s on the right hand side. The notation for the decoder g(\mu, \sigma, z) is also confusing because the decoder is supposed to map z to something in s space (see my first bullet point). The notation g(f(s), z) is particularly confusing because it is not consistent with the other notation that you use (which I mentioned in the sentence before). Usually, f(s) represents an element in latent space and is fed through the decoder to yield something in s space---so I don't understand why the decoder receives both f(s) and z as an argument.

- You talk about optimization objectives, then please specify what the optimization arguments are---this is not clear from the description given. For example, in Equation (1) the optimization arguments seem to be both the recognition (encoder) and the generative (prior over latents plus decoder) model parameters? In Equation (4), the optimization argument seems to be the latent variable z?

Given all the comments above, it is pretty obvious that the paper in its current form simply does not adhere to scientific standards for technically reporting machine learning algorithms in a proper way. So I clearly still vote for rejection because of the lack in technical clarity. And yes, as I said, I would like to see pseudocode.

**Experience Assessment:**

I do not know much about this area.

**Review Assessment: Checking Correctness Of Derivations And Theory:**

I assessed the sensibility of the derivations and theory.

**Review Assessment: Checking Correctness Of Experiments:**

I assessed the sensibility of the experiments.

**Review Assessment: Thoroughness In Paper Reading:**

I made a quick assessment of this paper.

---

> ### Author Response · Authors · 2019-11-10
> **Response to Reviewer 1**
>
> Before we answer your individual questions, we would like to emphasize two points.
> 1. The proposed method is a visualization/analysis method for RL agents. This means that it is inherently necessary to show visual results and explain them to the reader.
>
> 2. Even though it is in general hard to quantify visualization methods, we scientifically and quantitatively evaluate the quality of the reconstruction in Table 1 and ablate the influence of the proposed loss terms. In Table 2 and the histograms presented in Figure 8, we then verify that the whole pipeline is able to generate novel states of interest, while still covering the training distribution.
>
> > What is z in Section 2.1?
> In VAEs the reparametrization trick is used to sample from the predicted $\mu$ and $\sigma$. To this end, one samples a (e.g. Gaussian) vector $z$ and and uses it to generate a sample from the distribution $\mu + z\sigma$. We have added further clarifications to the paper.
>
> > How do the objectives in Section 2.1 and Section 2.2 relate to each other, i.e. how does the algorithm operate? Some pseudocode would be really helpful here.
>
> Section 2.1 describes how to train a generative model for the states in the environment. In a naive formulation the generative model would only be trained to reconstruct states from the environment. However, since the model will not be perfect, we have to encourage the generator to predict states that the agent can understand. This is what the proposed loss (1) for the generator is doing.
>
> Section 2.2 uses the (now trained) generative model to search for states of interest to the agent. We optimize the latent space of the generative model, minimizing the proposed target functions. You could interpret the generative model as a regularizer on the state space (which in this case is images). This regularization is needed since the agent has not seen out-of-domain images during training, thus directly optimizing pixels usually diverges into unwanted “adversarial” states. (See Figure 3)
>
> If the explanations above have not clarified the issue, please just let us know and we will gladly add pseudocode to the paper.
>
> > What are the target functions S^+, S^- and S^\pm in Section 2.3?
> Thank you. We have added their definition to the updated paper.
>
> > What is the difference in the KL-regularizer mentioned in the text below Equation (3) and in Equation (5)?
> The KL term in equation (1) is needed to train a variational autoencoder - which is the base for our generative model. We use it then also when optimizing the latent space (5) for states of interest as it prevents the samples to move too far from the Gaussian that the VAE was trained with.
>
> > Equation (2) is unclear
> We realize now, that attempting to keep the notation clean in Eq. (2) has made it less clear. The weighting of the reconstruction loss happens per pixel. The weight is the normalized gradient of the policy. We have rewritten the equation to now sum over individual pixels and hope that this resolves the confusion.
>
> > I do not think that nine pages are required to deliver the message of the paper in its current form.
> As this paper is a visualization paper, we have included 18 images containing state visualizations. As those use a lot of space we find it justified to use 9 pages instead of 8.

---

### Official Review · AnonReviewer3 · 2019-10-19
**Official Blind Review #3**

**Rating:** 6

**Review:**

The authors propose learning a generative model of states to visualize the behavior of different RL agents. Given states s from the environment, a VAE is trained to reconstruct states s, with an added loss term encouraging the action of the agent to stay the same between the original s and reconstructed s. The L2 reconstruction loss is also weighted by an attentive loss term, based on saliency of the policy pi, to focus reconstruction on critical regions.

Once the VAE is learned, we learn a 2nd sampling network, which operates in the latent space of the embedding. This energy-based model aims to sample regions of state space that optimize some target function T. For example, the target function may be "Q-value of moving left", in which case the model should generate states s where moving left is expected to be high-value. They examine a number of target functions, for maximizing / minimizing the Q-value of different actions, maximizing the spread of Q-values (max_a Q(s,a) - min_a Q(s,a)), and other saliency approaches. Experiments on Atari and a 3D driving simulator demonstrate that visualized images are qualitatively reasonable, and the states generated aren't just doing nearest neighbor over states in the training set.

The approach seems reasonable and the experiments seem reasonable as well. The saliency-based VAE objective is new to me as well. However, I contest the claim that this is one of the first works visualizing and diagnosing RL agents. The number of papers focusing purely on RL agent visualization is fairly small, but many deep RL papers include visualization as part of their experimental results, and in fact the paper cites some of these directly (like Greydanus et al, 2017). There are also a number of unclear details I'd want clarified.

Specific comments
* When describing the VAE training loss in Eqn (1), the last term can just be KL(f(s), N(0, I_n)), writing out the prior is more confusing, especially because the KL is mentioned in the text.
* Why is the gradient saliency measured in L1 rather than L2 distance? Every distance measured in the paper besides this one uses L2. Also, it is unclear what d is in the denominator - I assume it is a sum over each dimension of the state space but this is never fully described.
* Does the action consistency loss require differentiability through L_a? The provided example of the argmax action for policy pi seems like a hard loss function to learn, and it's unclear what L2 diff between two different argmax actions means.
* For the driving simulator, is there a reason the simulator used is an in-house one, rather than an existing driving simulator like CARLA?
* I buy the results showing the VAE learns to generate novel states. However, are there examples of novel states that drive insights that couldn't be found by doing nearest-neighbor over the training set? My thinking here is that you train E(x) the same way as before, except instead of x representing the noise passed to the VAE, you have x represent a non-parametric distribution that samples states from the replay buffer with some probability. For example, the paper argues they learned their Seaquest agent doesn't model the oxygen meter well, but was learning the generative model necessary to learn this?
* Section 2.3 (Target Functions) mentions target function S+, S-, S+-, none of which seem to be defined or mentioned in the main text.
* For prior visualization work, there is some related work from the adversarial RL literature (https://arxiv.org/pdf/1905.10615.pdf for a recent example), since adversarially attacking a policy tends to expose features that policy cares about. For examining failure states in particular, there is also https://arxiv.org/pdf/1812.01647.pdf which tries to identify catastrophic failures that are rare in the dataset.
* At a style level, I would not describe this paper as specifically a visualizing weaknesses paper - instead it is more like a framework to learn to generate states that satisfy some predicate of the Q-function (as noted by experiments that try to identify critical states, especially positive states, etc.), and I would consider renaming the paper accordingly.

Overall I feel this paper is very borderline but I'll round to weak accept.

Edit: I've read the author reply and other reviews. I don't plan to change my score. I thank the authors for clarifying some of the notational issues, and still believe it would be interesting to look at nearest-neighbor style baselines. If the environment is truly as complicated as stated, then these baselines should be very clearly bad and make a better case for the proposed contribution.

**Experience Assessment:**

I have published one or two papers in this area.

**Review Assessment: Checking Correctness Of Derivations And Theory:**

I assessed the sensibility of the derivations and theory.

**Review Assessment: Checking Correctness Of Experiments:**

I carefully checked the experiments.

**Review Assessment: Thoroughness In Paper Reading:**

I read the paper thoroughly.

---

> ### Author Response · Authors · 2019-11-10
> **Response to Reviewer 3**
>
> Thank you for your insightful comments and feedback! In the upcoming days, we have uploaded a revised version of the paper incorporating your and the other reviewers’ feedback.
>
> > I contest the claim that this is one of the first works visualizing and diagnosing RL agents
> We have toned down the claim for being one of the first RL diagnosis methods.
>
> > KL instead of prior in VAE training loss.
> Agreed. We have fixed this.
>
> > Why is the gradient saliency measured in L1 rather than L2 distance? What is d?
> With gradient saliency, one usually favors L1 over L2 since L2 tends to be very sharp and would only select very few pixels. d is the number of pixels - thank you, we have added this to the paper.
>
> > Does the action consistency loss require differentiability through L_a? Argmax seems hard to learn.
> We use $A(s)$ instead of $\pi(s)$ exactly to avoid this differentiability problem. In the given example $A$ would be the softmax logits predicted by the agent.
>
> > For the driving simulator, is there a reason the simulator used is an in-house one, rather than an existing driving simulator like CARLA?
> At the time that we did this work, CARLA wasn’t as mature as it is now and we wanted to explore some very specific scenarios involving pedestrians. We had already been creating our own car simulation environment to explore these ideas. The experiment could be reproduced in any arbitrary car environment with sufficient capabilities to animate pedestrians.
>
> > I buy the results showing the VAE learns to generate novel states. However, are there examples of novel states that drive insights that couldn't be found by doing nearest-neighbor over the training set?
>
> In Figure 8 we show that the states we generate are novel _and_ fulfil the target objectives. However, determining if they are problematic depends on external or human interpretation.
> Our method is motivated by real-world applications such as autonomous vehicles; where the environment is so complicated that it is impossible to obtain training data that densely samples critical regions of the environment distribution. In these situations with rich and complex environment distributions it becomes more important to have a generative model that is able to produce unseen states that are potentially problematic as well as consistent with a rich and complicated environment distribution. In this setting, a nearest neighbor search in the training set would not be able to retrieve the desired states due to the complexity of the environment’s distribution.
>
> > Section 2.3 (Target Functions) mentions target function S+, S-, S+-, none of which seem to be defined or mentioned in the main text.
>
> Thank you. The S target functions are the positive/negative sum of all Q-values. We had accidentally commented out the definition and added it back now.
>
> > adversarially attacking a policy tends to expose features that policy cares about.
> Thank you for these interesting references. We have discussed them in the Related Work section. The fact that it is easily possible to adversarily manipulate an agent motivates the use of our generative model further. We are not looking for adversarial states that make the agent fail - instead we would like to find semantically meaningful flaws. Constraining the search space to the latent representation of a generative model helps in this regard (Section 4.6).

---

### Official Review · AnonReviewer2 · 2019-10-24
**Official Blind Review #2**

**Rating:** 8

**Review:**

This paper proposes a new visualization tool in order to understand the behavior of agents trained using deep RL. Specifically, they train a generative model of game states, and then optimize an energy-based distribution over state embeddings according to some target function, and then by sampling from the resulting distribution they create a diverse set of realistic states that score highly according to the target function. They propose a few target cost functions, which allow them to optimize for states in which the agent takes a particular action, states which are high reward (worst Q-value is large), states which are low reward (best Q-value is small), and critical states. They demonstrate results on Atari games as well as a simulated driving environment.

I enjoyed this paper: the proposed method is straightforward, and the experiments are well-done and demonstrate the potential of the method, and for that reason I’m recommending an accept.

If I had to name a fault with the paper, it is that the interpretation of the results depends quite strongly on human judgment: we are asked to look at particular images and then told how to interpret them. It would be both more compelling and more interesting to use the results to *fix* problems detected in the agents. For example, Section 4.4 suggests that the Seaquest agent has not learned that it should surface when the oxygen is low. Having done this analysis, can we then fix the problem? Perhaps it would work to sample states according to the T+- objective, and then train the agent to take the “up” action in such situations, perhaps interspersed with regular RL / training on the replay buffer to avoid catastrophic forgetting. If the problem could be fixed, that would be strong evidence for the utility of the method.

Minor questions:

Can you explain why T+-(q) = T-(q) - T+(q) incentivizes “situations in which one action is of very high value and another is of very low value”? It is not immediately obvious to me why this should be true, just from the definition.

Typos:

Second contribution: “interstingness” --> “interestingness”
Section 3.2: “In the next section, we will show how to overcome these difficulties.” I assume this is referring to what is now Section 2?
Please organize the figures better in relation to the text (e.g. Fig. 5 should be after Figs. 6 and 7).

**Experience Assessment:**

I have read many papers in this area.

**Review Assessment: Checking Correctness Of Derivations And Theory:**

N/A

**Review Assessment: Checking Correctness Of Experiments:**

I assessed the sensibility of the experiments.

**Review Assessment: Thoroughness In Paper Reading:**

I read the paper at least twice and used my best judgement in assessing the paper.

---

> ### Author Response · Authors · 2019-11-10
> **Response to Reviewer 2**
>
> Thank you for your insightful comments!
>
> > Use the intuitions to “fix” the agent.
> We agree that it would be great if we could systematically show that we can use this method to fix problems in an agent. However, in the Seaquest game for example the T+- objective does not exclusively select low oxygen states. There are other states where enemies or power-ups are close that also trigger the same objective. Thus, always encouraging the up-action might not be the right choice.
> However, with the insight that oxygen might be a problem one could either design a specific environment to train the agent in that emphasizes this hazard, or write an oxygen detector tool that supplies this information to the agent to encourage it to account for it in the policy.
> In a real world setting, for example when analyzing a driving agent, one would need to search a large database of recorded driving footage for data that is similar to the visualized failure state. With the visualization a search can be much more directed - and semantic - evaluating the agent for every state in the database.
> Overall, we believe that fixing agents is very task (and agent) specific and goes beyond the scope of this paper - which is to find and understand flaws.
>
> > Why does T+- optimize for situation where one action is of very high value and another is of very low value?
> The intuition might be easier to follow if you think of the non-soft version of T-: max_i(q_i). It returns simply the highest Q-value. T+ in turn selects the lowest Q-value. Thus, T+-(q) = T-(q) - T+(q) optimizes for a high difference between highest and lowest Q-value. These are actions where the agent sees both a high and a low future (discounted) reward which we find is a good proxy for critical situations. These are states where the choice of action matters the most.
>
> > Typos
> We will fix all typos and upload a revised version of the paper with reordered figures in the coming days.

---

> > ### Comment · AnonReviewer2 · 2019-11-15
> > **Thanks!**
> >
> > Thanks for the response.
> >
> > While I agree that fixing agents is specific to the problem at hand, even a task-specific fix would nonetheless be a much more compelling then just visualizations. I would treat it as part of the evaluation, not as part of the contribution.
> >
> > I agree with R1 that your VAE notation is confusing. I would recommend renaming $z$ to $\epsilon$, and defining $z = \mu + \sigma \epsilon$, so that $\epsilon$ is the random variable used in the reparameterization trick, which is the more standard notation. Then, as R1 expects, $z$ would be the result of sampling from the distribution given by $f(s)$, and $g$ would be a function only of $z$. For an example, see the notation in this blog post: https://jaan.io/what-is-variational-autoencoder-vae-tutorial/ (in particular the section on the reparameterization trick).
> >
> > Having read the other reviews and author response, my score remains the same (though I do encourage the authors to fix the VAE notation).

---

### Decision · Program_Chairs · 2019-12-19

**Decision:**

Accept (Poster)

**Comment:**

This paper proposes a tool to visualizing the behaviour of deep RL agents, for example to observe the behaviour of an agent in critical scenarios. The idea is to learn a generative model of the environment and use it to artificially generate novel states in order to induce specific agent actions. States can then be generated such as to optimize a given target function, for example states where the agent takes a specific actions or states which are high/low reward. They evaluate the proposed visualization on Atari games and on a driving simulation environment, where the authors use their approach, to investigate the behaviour of different deep RL agents such as DQN.

The paper is very controversial. On the one hand, as far as we know, this is the first approach that explicitly generates states that are meant to induce specific agent behaviour, although one could relate this to adversarial samples generation. Interpretability in deep RL is a known problem and this work could bring an interesting tool to the community. However, the proposed approach lacks theoretical foundations, thus feels quite ad-hoc, and results are limited to a qualitative, visual, evaluation. At the same time, one could say that the approach is not more ad hoc than other gradient saliency visualization approaches, and one could argue that the lack of theoretical soundness is due to the difficulty of defining good measures of interpretability and that apply well to image-based environments.

Nonetheless, this paper is a step in the good direction in a field that could really benefit from it.